# Performance Improvement of Underwater Continuous-Variable Quantum Key Distribution via Photon Subtraction

**DOI:** 10.3390/e21101011

**Published:** 2019-10-17

**Authors:** Qingquan Peng, Guojun Chen, Xuan Li, Qin Liao, Ying Guo

**Affiliations:** 1Jiangsu Key Construction Laboratory of IoT Application Technology, School of IoT Engineering, Wuxi Taihu University, Wuxi 214064, China; weiyideai520@gmail.com (Q.P.); chengj@wxu.edu.cn (G.C.); 2School of Automation, Central South University, Changsha 410083, China; 3School of Economics and Management, Jingzhou Institute of Technology, Jingzhou 434020, China; maoyun3106@sina.com; 4College of Computer Science and Electronic Engineering, Hunan University, Changsha 410082, China

**Keywords:** continuous variable quantum key distribution, photon subtraction operation, underwater channel

## Abstract

Considering the ocean water’s optical attenuation is significantly larger than that of Fiber Channel, we propose an approach to enhance the security of underwater continuous-variable quantum key distribution (CVQKD). In particular, the photon subtraction operation is performed at the emitter to enhance quantum entanglement, thereby improving the underwater transmission performance of the CVQKD. Simulation results show that the photon subtraction operation can effectively improve the performance of CVQKD in terms of underwater transmission distance. We also compare the performance of the proposed protocol in different water qualities, which shows the advantage of our protocol against water deterioration. Therefore, we provide a suitable scheme for establishing secure communication between submarine and submarine vehicles.

## 1. Introduction

Quantum communication is a new type of communication method and quantum key distribution (QKD) [1,2,3,4] is part of it. Underwater communication is essential in modern communications since it relates to the interaction of information between various devices underwater. The traditional way of underwater communication is to use underwater acoustic technology and its shortcomings are obvious, such as high extension, low bandwidth and security issues [5]. To solve these problems, research has found that undersea optical communication can effectively improve communication bandwidth and reduce delay. Therefore, underwater optical communication has attracted attention and has gradually become the new favorite of marine communication in the modern era. Recent research based on underwater optical communication has not only improved data bandwidth but also reduced the bit error rate [6,7]. Although there is an increase in security compared to underwater acoustic technology, there are still some fatal security holes [8]. Fortunately, CVQKD protocol has proven to be unconditionally safe in theoretical research, which is determined by its unique physical properties [9,10,11]. The introduction of CVQKD can better solve the problem of underwater communication security. Currently, both Fiber Channel and free space are used to achieve unconditionally secure quantum communication [12,13]. However, CVQKD’s secure key rate calculations are generally complex. Because in open quantum systems, eigenvalues not only provide energy but also the lifetimes of the states of the system [14,15,16].

The effective transmission distance of underwater CVQKD is significantly weaker than that of both fiber and air channels but it can be applied to the height safety areas. For example, underwater detection equipment can safely transmit information to an underwater platform [17]. In this way, encrypted information can be securely shared between the submarine and the device or device and device when working together. However, the original underwater CVQKD protocol is affected by the short transmission distance, which is always difficult to achieve. We propose a novel method that uses photon subtraction operation [18,19,20] to improve the performance of underwater CVQKD and extend its effective transmission distance. This potentially allows one to establish the safest and largest underwater communication network [5].

The purpose of this paper is to improve the performance of underwater CVQKD and analyze its performance based on the transmittance of seawater. At present, there have been many experiments to prove that apply photon subtraction operation can improve the entanglement between quantum, prolong the transmission distance of CVQKD protocols using two-mode squeezed vacuum (TMSV) states [21,22,23,24]. Generally, entanglement generated via a beam splitter when an excited coherent state is injected on Alice input mode and vacuum state is injected on the other one [25,26]. It is widely used in entanglement-based (EB) scheme or prepare- and-measure (PM) scheme, even in protocols that use coherent states. In the case of underwater CVQKD, the effect of CVQKD is not only the entanglement between the quantum but also the photon intensity, where photon intensity depends on the degree of squeezing [27,28]. However, the photon subtraction operation is a non-Gaussian operation [29,30,31], which will allow us to simulate the ideal photon subtraction operation by continuously adjusting the parameters and obtaining the best performance of the underwater CVQKD transmission. Another advantage of photon subtraction is that it does not require complex physical operations. The simulation results show that the photon subtraction operation can effectively extend the transmission performance of underwater CVQKD, resulting in the optimal performance by subtracting one photon.

This paper is organized as follows: In Section 2, we suggest the design of the CVQKD system based on photon subtraction and demonstrate the how seawater affects light. In Section 3, we perform a security analysis of the modified CVQKD protocol. In Section 4, we show the simulation results of the secret key rates of underwater CVQKD via photon subtraction. Finally, we draw a conclusion in Section 5.

## 2. Underwater CVQKD via Photon Subtraction

### 2.1. System Description

In this section, we describe the underwater CVQKD scheme with photon subtraction. We find that photon subtraction can enhance the entanglement of TMSV state, thereby improving CVQKD scheme. In addition, as a quantum information transmission medium, water has a large attenuation compared with fiber or atmosphere, therefore we consider the water transmittance of CVQKD scheme in what follows.

The CVQKD system with photon subtraction is depicted in Figure 1. TMSV can be used for generating two modes *A* and *B*, where *A* is the annihilation operator, denoted as $\left\{\hat{a},{\hat{a}}^{†}\right\}$ and *B* is the creation operator, denoted as $\left\{\hat{b},{\hat{b}}^{†}\right\}$. *A* and *B* exist in the equation $\left[\hat{a},{\hat{a}}^{†}\right]=\left[\hat{b},{\hat{b}}^{†}\right]=1$. The resulting TMSV state can be written as
(1)$$|TMSV\rangle =\sqrt{1-\lambda^{2}}\sum_{n=0}^{\infty }\lambda^{n}|n,n\rangle ,$$
where $\lambda \in \left[0,1\right),|m,n\rangle ={|m\rangle }_{A}⊗{|n\rangle }_{B}$ and ${\left\{|n\rangle \right\}}_{n\in N}$ denotes the fock state. In addition, the TMSV state can be represented by the variance *V*, where
(2)$$V=\frac{1+\lambda^{2}}{1-\lambda^{2}}.$$

Alice performs heterodyne detection and photon subtraction operations on modes *A* and *B* generated by TMSV, respectively. The mode *B* will be transformed into mode $B_{2}$ after the photon subtraction operation. Then, Alice sends the quantum signals mode $B_{2}$ to the receiver Bob via a water channel with transmittance $T_{c}$ and excess noise ξ. Consequently, Bob receives the mode $B_{3}$, where the total channel-added noise referred to the channel input is expressed in shot noise units as
(3)$$X_{line}=\left(1-T_{c}\right)/T_{c}+\xi .$$
When the above process is completed, Alice and Bob use the accepted data to parameter estimation, information coordination and privacy amplification.

Since Alice can decide whether or not to approve or deny each data after Bob completes the measurement, the rejected data can be considered as the bait data to enhance the security of the former non-Gaussian protocol [32].

### 2.2. Photon Subtraction

On Alice’s side, the EB scheme of CVQKD via photon subtraction operation is shown in Figure 2. Mode *B* passes through the beam splitter with a transmittance of *T* to obtain modes $B_{1}$ and $B_{2}$. Then the state $\rho_{AB_{1}B_{2}}$ can be expressed by
(4)$$\rho_{AB_{1}B_{2}}=U_{BS}\left[|TMSV\rangle \langle TMSV|⊗|0\rangle \langle 0|\right]U_{BS}^{†}.$$ Mode $B_{1}$ will be measured by positive operator-valued measure (POVM) measurements $\left\{{\hat{\prod }}_{0},{\hat{\prod }}_{1}\right\}$. Only when the POVM element ${\hat{\prod }}_{1}$ clicks modes *A* and $B_{2}$ will be kept. The photon subtracted TMSV state is given by
(5)$$\rho_{AB_{2}}^{{\hat{\prod }}_{1}}=\frac{tr_{B_{1}}\left({\hat{\prod }}_{1}\rho_{AB_{1}B_{2}}\right)}{tr_{AB_{1}B_{2}}\left({\hat{\prod }}_{1}\rho_{AB_{1}B_{2}}\right)},$$
where $tr_{x}\left(\cdot \right)$ is the partial trace of the multimode quantum state and the success probability of subtracting k photons is represented by $tr_{AB_{1}B_{2}}\left({\hat{\prod }}_{1}\rho_{AB_{1}B_{2}}\right)$, given by
(6)$$\begin{array}{cc} P_{\left(k\right)}^{{\hat{\prod }}_{1}}= & tr_{AB_{1}B_{2}}\left({\hat{\prod }}_{1}\rho_{AB_{1}B_{2}}\right)\\ = & \left(1-\lambda^{2}\right)\sum_{n=k}^{\infty }\lambda^{2n}C_{n}^{k}T^{n-k}{\left(1-T\right)}^{k}\\ = & \frac{\left(1-\lambda^{2}\right){\left(1-T\right)}^{k}\lambda^{2k}}{{\left(1-T\lambda^{2}\right)}^{k+1}}, \end{array}$$
where $C_{n}^{k}$ is combinatorial number and its relationship with the transmittance of Alice’s $BS_{1}$ is shown in Figure 2. After passing the Alice’s $BS_{1}$, the state $\rho_{AB_{2}}^{{\hat{\prod }}_{1}}$ entanglement degree increases via the photon subtraction operation, while worth noticing that the state is not Gaussian anymore.

Alice perform heterodyne detection and Bob uses homodyne detection, which is convenient to implement in experimentation. Suppose $\gamma_{AB_{2}}^{\left(k\right)}$ represents the covariance matrix of $\rho_{AB_{2}}^{{\hat{\prod }}_{1}}$ and it can be given by
(7)$$\begin{array}{c} \gamma_{AB_{2}}^{\left(k\right)}=\left(\begin{array}{cc} a\prod & c\sigma_{z}\\ c\sigma_{z} & b\prod \end{array}\right), \end{array}$$
where ∏ = dialog(1,1) and $\sigma_{z}$ = dialog(1,-1) with
(8)$$\begin{array}{cc} a= & \frac{T\lambda^{2}+2k+1}{1-T\lambda^{2}},\\ c= & \frac{2\sqrt{T}\lambda \left(k+1\right)}{1-T\lambda^{2}},\\ b= & \frac{T\lambda^{2}\left(2k+1\right)+1}{1-T\lambda^{2}}. \end{array}$$

In the CVQKD system, Alice uses the photon subtracted TMSV state as the source and she will perform heterodyne detection on mode A. As shown in Figure 2. Alice needs to record the measurement results of each TMSV from a single-photon detector (click: keeping this state; no-click: not keeping this state). Alice will reveal this extra data it to Bob, after Bob measures the mode $B_{3}$.

### 2.3. Seawater Channel

When Alice and Bob exchange information, Alice and Bob are assumed to be peer-to-peer. In this case, the fact that light is scattered in the forward direction has a positive impact on underwater communication because it is the amount of such scattered light source that reaches the underwater receiver from the perspective of the link budget perspective. Figure 3 shows the beam-spread model diagram, where $D_{src}$ and $D_{rec}$ are photon emission source and photon receiver, respectively. As shown in the model diagram, the optical source is located at depth and perpendicular to the receiving plane. When photons emitted by Alice flow through seawater to the Bob, the photon scattering randomly hits Bob’s receiving plane instead of the original beam axis center point. We find that the applicable receiving radius *r* becomes larger as the depth *D* increases.

Figure 4 shows Bob receiving photons emitted by Alice propagate 0 m, 6 m and 12 m in pure seawater. The brighter the color in the picture (bright yellow), the greater the light intensity and the darker the color (dark blue), the weaker the light intensity. The photons sent by Alice is propagate in pure seawater and its photon intensity shows a significant weakening trend, which by vertical height from Figure 4a–c. It is worth noting that the initial photon intensity is related to the squeezed TMSV [27]. The photons received by Bob are in the form of scattering and the scattering is more pronounced as the distance increases. As can be seen from Figure 4d–f, the spot gradually becomes smaller. Undoubtedly, this phenomenon is mainly assigned to the absorption and scattering of light by seawater. At the same time, Figure 4 also shows that this trend will provide Eve with the opportunity to get more information when transmitting over long distances.

Obviously, both the model map and the photon intensity map show that the seawater can hugely influent photon’s transmission. Next, in view of the above phenomenon, we build a closed system where the dissipation is neglected, study the effect of photon subtraction operation on underwater CVQKD and give the derivation for calculating the transmittance of photons in seawater.

Light travels harder in seawater than both fiber and atmospheric channels, since there are two processes affect light propagation in seawater, which called absorption and scattering respectively. Although we all know that photon transmission paths are straight lines, the transmission path of light can be changed especially in the seawater, as the photons many interact with certain particles. This phenomenon is called scattering. On the other side, photon energy loss caused by the interaction of photons and particles is called absorption, which is an irreversible thermal process. The superposition of the above process becomes a description of the beam attenuation coefficient $Q_{at}$ in seawater, including absorption coefficient $Q_{ab}$ and scattering coefficient $Q_{sc}$. Then, both of the above coefficients are affected by wavelength ι and chlorophyll concentration C. The total attenuation coefficient $Q_{at}$ is defined as the sum of $Q_{ab}$ and $Q_{sc}$,
(9)$$\begin{array}{cc} Q_{ab}\left(\iota \right)= & \left[a_{w}\left(\iota \right)+0.06a_{c}\left(\iota \right)C^{0.65}\right]\left[1+0.2e^{-0.014(\iota -440)}\right],\\ Q_{sc}\left(\iota \right)= & 0.3\frac{550}{\iota }C^{0.62},\\ Q_{at}\left(\iota \right)= & Q_{ab}\left(\iota \right)+Q_{sc}\left(\iota \right), \end{array}$$
where $a_{w}\left(\iota \right)$ is the absorption coefficient in seawater and $a_{c}\left(\iota \right)$ is the specific absorption coefficient of chlorophyll, both based on statistical derivation.

Assuming that the underwater channel is a linear attenuation model. The transmittance of seawater can be calculated by
(10)$$T_{sea}=exp\left(-Q_{at}\left(\iota \right)D\right),$$
where *D* is the depth.

## 3. Performance Analysis

In this section, we first derive the secret key rate for underwater CVQKD protocol via photon subtraction. Then, the parameters involved in the secret key rate are estimated for the improvement of the underwater CVQKD system.

### 3.1. Secret Key Rate

After the quantum signal sent by Alice is received by Bob, the final covariance matrix between them can be described as
(11)$$\begin{array}{c} \gamma_{AB_{3}}^{\left(k\right)}=\left(\begin{array}{cc} a^{\prime }\prod & c^{\prime }\sigma_{z}\\ c^{\prime }\sigma_{z} & b^{\prime }\prod \end{array}\right) \end{array},$$
where $a^{\prime }=a$, $b^{\prime }=T_{sea}\left(b+X_{line}\right)$ and $c^{\prime }=\sqrt{T_{sea}}c$. Then $\gamma_{AB_{3}}^{\left(k\right)}$ can be rewritten as
(12)$$\begin{array}{c} \gamma_{AB_{3}}^{\left(k\right)}=\left(\begin{array}{cc} a\prod & \sqrt{T_{sea}}c\sigma_{z}\\ \sqrt{T_{sea}}c\sigma_{z} & T_{sea}\left(b+X_{line}\right)\prod \end{array}\right) \end{array}.$$

Assuming that Alice and Bob perform heterodyne detection and homodyne detection respectively, the resulting secret key rate in reverse reconciliation scenario can be given by
(13)$$K^{Hom}=P_{\left(k\right)}^{{\hat{\prod }}_{1}}\left[\beta I_{\left(A:B\right)}^{Hom}-S_{\left(E:B\right)}^{Hom}\right],$$
where β is the reconciliation efficiency, Hom is homodyne detection and I(A:B) denotes the mutual information between Alice and Bob, and S(E:B) is the mutual information between Eve and Bob. In detail, Alice and Bob have achieved mutual information
(14)$$\begin{array}{c} I_{\left(A:B\right)}^{Hom}=\frac{1}{2}log_{2}\frac{V_{A}}{V_{A|B}^{Hom}}, \end{array}$$
where $V_{A}=\left(a^{\prime }+1\right)/2$, $V_{B}=b^{\prime }$ and $V_{A|B}^{Hom}=V_{A}-c^{\prime 2}/\left(2V_{B}\right)$. Eve and Bob have mutual information
(15)$$\begin{array}{c} S_{\left(E:B\right)}^{Hom}=G\left(\frac{f_{1}-1}{2}\right)+G\left(\frac{f_{2}-1}{2}\right)-G\left(\frac{f_{3}-1}{2}\right), \end{array}$$
where the function $G\left(x\right)=\left(x+1\right)log_{2}\left(x+1\right)-xlog_{2}x$ and the eigenvalues $f_{i(i=1,2,3)}$ are expressed as
(16)$$\begin{array}{cc} f_{1,2}^{2}= & \frac{1}{2}\left(\Delta \pm \sqrt{\Delta^{2}-4D^{2}}\right),\\ f_{3}^{2}= & a^{\prime }\left(a^{\prime }-\frac{c^{\prime 2}}{b^{\prime }}\right), \end{array}$$
with the symbols given by
(17)$$\begin{array}{cc} \Delta & =a^{\prime 2}+b^{\prime 2}-2c^{\prime }2,\\ D & =a^{\prime }b^{\prime }-c^{\prime 2}. \end{array}$$ Nothing that Eve’s best attack for this non-Gaussian protocol is still an open question, we here mainly consider the classic attack which called entanglement-clone attack. In the following, we will focus on effect of the non-Gaussian operation involved photon subtraction on the seawater-related quantum channel.

### 3.2. Parameters Estimation

The effect of the light in seawater can be described with absorption coefficient $Q_{ab}\left(\iota \right)$ and scattering coefficient $Q_{sc}\left(\iota \right)$. The two coefficients sums are the attenuation coefficient of seawater $Q_{at}\left(\iota \right)$ and both have a relationship with the wavelength ι. Figure 5 shows that clear ocean water absorbs light minimally when the wavelength is between 450 nm and 550 nm which coincides with the existence of a blue-green optical window in seawater.

According to the above-mentioned analysis, 520 mm light is chosen to demonstrate the influence of propagation distance on the performance of the underwater CVQKD system via the non-Gaussian operation. The coefficient values $Q_{ab},Q_{sc}$ and $Q_{at}$ are shown in Table 1 for simple comparison [33,34].

After Alice performs the *k*-photon subtraction operations, the probability of success can be determined by the transmittances *T* of $BS_{1}$, as shown in Figure 6. It shows the probability of success for Alice using $BS_{1}$ as function of transmittances *T*. The lines from top to bottom represent the probabilities of success for one-photon subtraction, two-photon subtraction, three-photon subtraction and four-photon subtraction, respectively. We find that as the number of photons increases, the probability of success decreases. The best of success probability is 25% when one-photon subtraction is performed by Alice.

## 4. Simulation

In this section, we simulate the secret key rate of underwater CVQKD via photon subtraction. The secret key rate of CVQKD is affected by various parameters when calculating. Therefore, in order to reflect the actual situation more realistically, we performed dynamic analysis on different parameters to obtain different secret key rate simulation results. It includes different *k*-photon subtraction operation, different water quality and different transmittance *T* of $BS_{1}$. These simulation results will prove that our proposed an approach to the security of underwater CVQKD is useful.

The performance of CVQKD under pure seawater via photon subtraction are shown in Figure 7. the results show that both the original CVQKD system and the improved CVQKD involving photon subtraction operation can safely perform quantum communication when the depth is lower than 71.7 m. Fortunately, Alice and Bob can still obtain positive secret key after performing one-photon subtraction or two-photon subtraction when the depth is greater than 72.1 m. We find that the maximum transmission distance using one-photon subtraction operation can be 123.1 m for the one-photon subtraction-involved seawater CVQKD system.

In the photon subtraction operation, the transmittance *T* of the $BS_{1}$ is of very important. Figure 8 respectively shows the secret key rates of CVQKD via one-photon subtraction (a) and two-photon subtraction (b) with different *T* range from 0–1 and different depth range from 0–140 m. When the transmission depth is short, the smaller the transmittance (T>0.5), the larger the secret key rate value. This is mainly due to the increase in the degree of entanglement between the quantum so that even if the transmittance is small, Bob can receive a strong quantum signal in the case of short distances. When the vertical distance between Alice and Bob is increasing, an appropriate increase *T* in transmission can extend the transmission distance while ensuring safe communication. Regardless of the one-photon subtraction operation or the two-photon subtraction operation, the performance situation is similar. However, since the success probability of two-photon subtraction operation is less than that of one-photon subtraction operation, the transmission distance of CVQKD via two-photon subtraction operation is outperformed by CVQKD via one-photon subtraction operation.

Figure 9 shows the comparison of original underwater CVQKD protocol and the improved underwater CVQKD with one-photon subtraction. Lines from top to bottom represent pure sea water (red), clear sea water (brown) and coastal sea water (blue), respectively, where the solid line is the original protocol and the dotted line is the proposed protocol. The results show that the maximum transmission distance of CVQKD can be improved after performing the photon subtraction operation for each quality of seawater. However, the maximum transmission distance significantly decreases as the quality of seawater deteriorates. For example, taking the improved protocol into account, the maximal transmission distance in pure seawater can reach 123.1 m, whereas the maximum transmission distance in coastal ocean seawaters is only 13.3 m.

## 5. Conclusion

We have suggested an improved underwater CVQKD by performing photon subtraction. In order to lengthen the transmission distance of the system, we deploy the photon subtraction-involved non-Gaussian operation in Alice’s operating procedures. We obtain the optimal absorption coefficient value under different seawater quality using the extensive data, although the probability of successful photon subtraction is affected by the transmittance of Alice’s BS. The simulation results show that the maximal transmission distance is remarkably increased after using photon subtraction. We also compare the performance of the proposed protocol in different water quality, which shows the advantage of our protocol against water deterioration. Moreover, the simulation parameters come from the reported experiments, which is very useful for our future research on building a global underwater secure communication network.

## Figures and Tables

**Figure 1 entropy-21-01011-f001:**
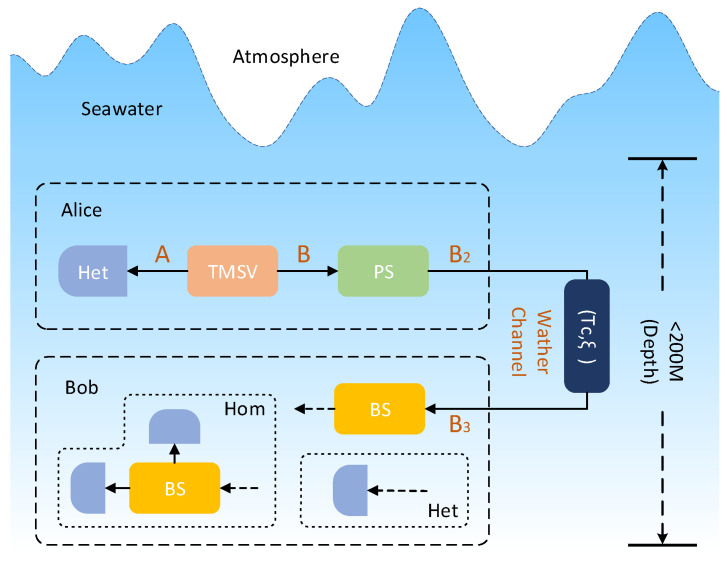
Underwater CVQKD via photon subtraction model diagram. Het: heterodyne detection; Hom: homodyne detection; PS: photon subtraction; BS: beam splitter; $T_{c},\xi$,: channel parameters operator.

**Figure 2 entropy-21-01011-f002:**
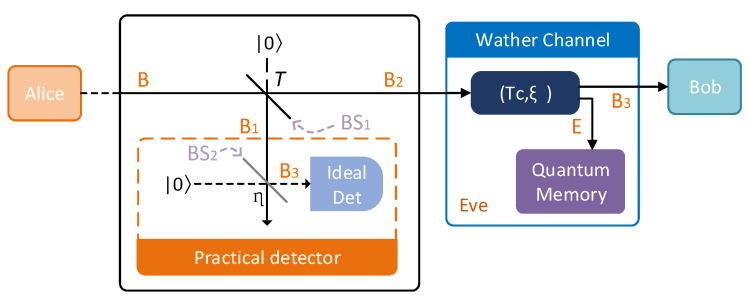
The EB scheme of CVQKD via photon subtraction operation. BS1,2: beam splitter; T,η: transmittance of BS1,2; $T_{c},\xi$,: channel parameters operator.

**Figure 3 entropy-21-01011-f003:**
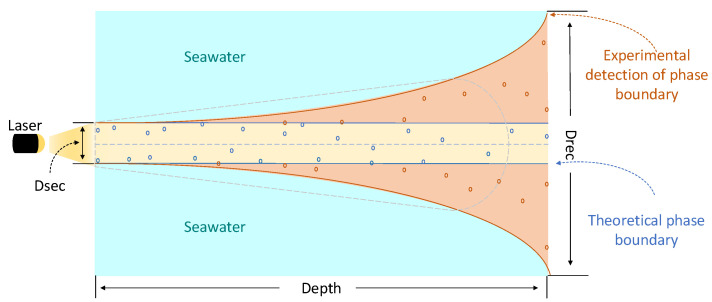
Beam-spread model diagram. $D_{src}$: Photon emission source; $D_{rec}$: Photon receiver.

**Figure 4 entropy-21-01011-f004:**
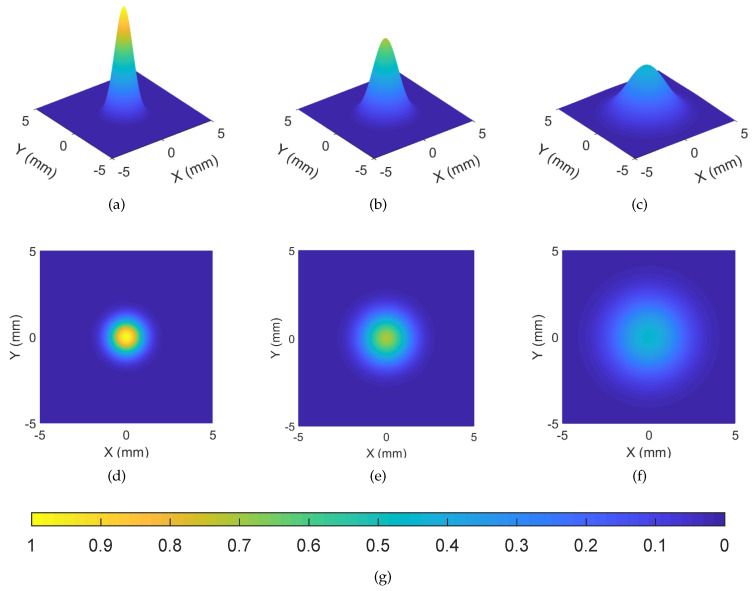
Simulating Bob receiving photons. (**a**,**d**) 0-m pure sea water; (**b**,**e**) 6-m pure sea water; (**c**,**f**) 12-m pure sea water; (**d**) 12m pure sea water; (**g**) Photon intensity level.

**Figure 5 entropy-21-01011-f005:**
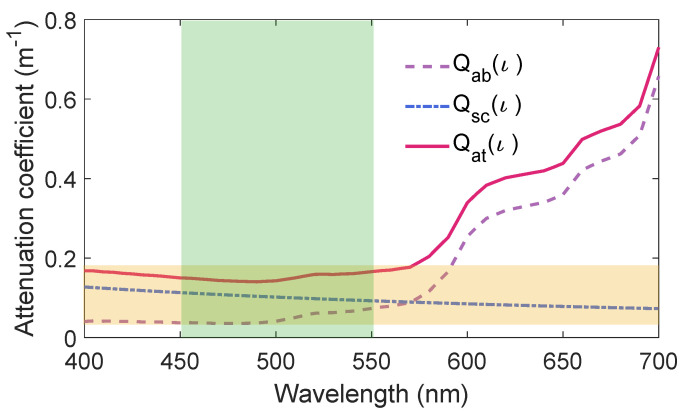
Clear ocean water attenuation as a function of wavelength.

**Figure 6 entropy-21-01011-f006:**
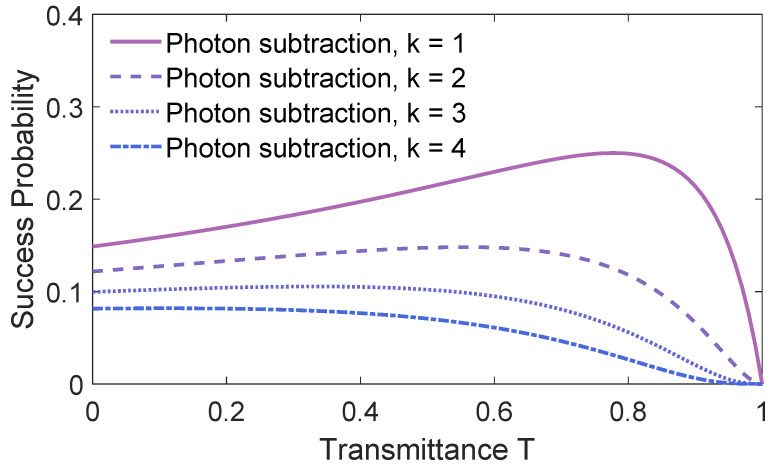
The success probability of subtracting *k* photons. The variance of TMSV is V=10.

**Figure 7 entropy-21-01011-f007:**
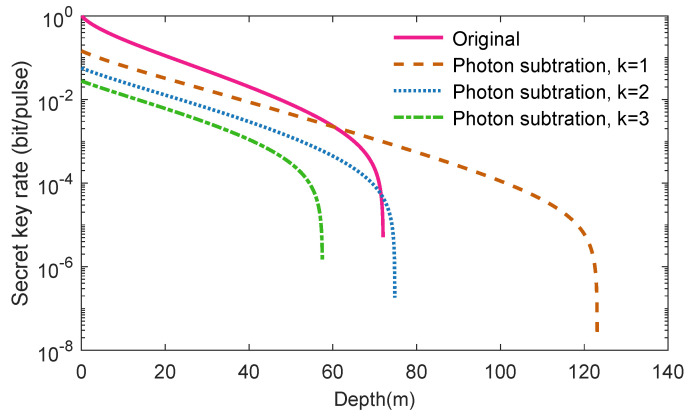
The secret key rate of the CVQKD under pure seawater via *k*-photon subtraction. Parameters are set to V=10,ξ=0.04,β=0.96 and $C_{c}=0.043$.

**Figure 8 entropy-21-01011-f008:**
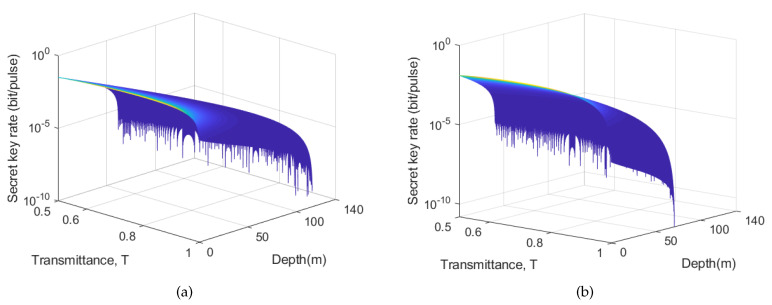
The secret key rate of the CVQKD under pure seawater. (**a**) One-photon subtraction operation; (**b**) Two-photon subtraction operation

**Figure 9 entropy-21-01011-f009:**
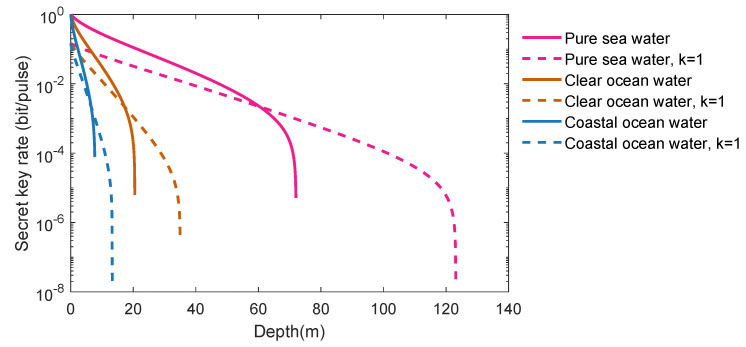
The secret key rate for the underwater CVQKD via one-photon subtraction.

**Table 1 entropy-21-01011-t001:** Coefficient values of typical water types at 520 nm.

Water Types	$Q_{\mathrm{ab}}\left(m^{-1}\right)$	$Q_{\mathrm{sc}}\left(m^{-1}\right)$	$Q_{\mathrm{at}}\left(m^{-1}\right)$
Pure sea water	0.0405	0.0025	0.043
Clear ocean water	0.114	0.037	0.151
Coastal ocean water	0.179	0.219	0.398

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
