# Peer review of "Performance Improvement of Underwater Continuous-Variable Quantum Key Distribution via Photon Subtraction"

_entropy, 2019, doi:10.3390/e21101011_

Round 1

Reviewer 1 Report

The manuscript concerns potentially interesting topics of quantum communication in real situations. In particular, the Authors have concentrated on the problem of the quantum key distribution in water media. The manuscript deals with the so-called ``hot topis’’ which deserve to be considered in articles. Unfortunately, the form of the paper should be considerably improved. It

exhibits a lack of rigorous scientific approach, starting from the language errors which should be amended. The Reader can find there many flaws and ambiguities. One can find in the manuscript sudden jumps from one formula to the next ones (for instance, see lines 85 and 86) without any explanation in which way the result was obtained. Not all elements of the schemes shown in Fig.2 were explained. The Authors discuss the quantities which are not properly defined and(or) described – for instance, see lines 113 and 114. The parameters A and B, which appear for the first time at line 61, are, according to the Authors’ explanations, annihilation operators, next they are called  ``models’’ or states (line 67). In general, the commentaries given by the Authors are unclear (for instance, see lines 66-70). The Authors stated that Fig.4 shows the photon distribution. However, they do not explain in which context they use the term ``distribution’’. One can see that x and y axes appearing there are scaled in millimeters, which suggests that some space dependence of the light intensity is shown there. Those are only examples of the flaws which can be found in the manuscript.

In my opinion, the article contains potentially valuable results that should be published, but the form of their presentation is far from the usually accepted standards, especially those of the Entropy journal. Therefore, I recommend to rewrite the manuscript in the proper way and resubmit it again. My recommendation is `` major revision’’.

Author Response

Thank you very much for your careful and constructive advices. Those comments are all valuable and very helpful for revising and improving our paper, and with the important guiding significance to our researches. We have tried our best to revise and improve the manuscript and made some changes in the manuscript. Please refer to the revised paper and following answers. Thank you again.

Reviewer 2 Report

The authors proposed a new scheme for secure communication between submarines.

They propose an approach to enhance the security of underwater continuous-variable

quantum key distribution. In this approach the quantum entanglement is enhanced by the photon subtraction at the emitter.

The subject is interesting but before giving any decision, revision is required. The following comments should be considered for the revised version:

The work seems to be done only in a closed system where the dissipation is neglected. The authors should explicitly clarify this in the paper. The physical effects in the quantum open system should discussed briefly in the introduction or before the conclusion, PRA 95, 022117 (2017); EPJD 68, 74 (2014); EPJD 69, 229 (2015) and references therein. are helpful and should be cited. The author should include more discussion on the quantum correlations and entanglement in general by giving some examples in physical systems, references EPJD 68, 191 (2015); Quant. Inf. Process. 12, 69 (2013); Phys.Rev. A 84, 053817 (2011): PRA 91, 032309 (2015) and references therein are helpful and should be cited. The usual method of Continuous-variable quantum key distribution should be introduced (adding a section after the introduction will increase the visibility of the paper)

Author Response

(The authors gave the same response as above.)

Round 2

Reviewer 1 Report

The submitted manuscript is an amended version of that already submitted to the Entropy journal.  In my opinion, the Authors answered satisfactorily for all doubts and questions from the Referee’s report. The new version of the article is considerably improved, and thus, I recommend the article for publication in its present form.

Reviewer 2 Report

The revised version is ok. I accept the paper.